# Optimizing the Microstructure and Mechanical Properties of Vacuum Counter-Pressure Casting ZL114A Aluminum Alloy via Ultrasonic Treatment

**DOI:** 10.3390/ma13194232

**Published:** 2020-09-23

**Authors:** Gang Lu, Pengpeng Huang, Qingsong Yan, Pian Xu, Fei Pan, Hongxing Zhan, Yisi Chen

**Affiliations:** National Key Laboratory of Light Alloy Processing Science and Technology, Nanchang Hangkong University, Nanchang 330063, China; 1703080503009@stu.nchu.edu.cn (P.H.); yanqs1973@nchu.edu.cn (Q.Y.); xupian@nchu.edu.cn (P.X.); 1503080503015@stu.nchu.edu.cn (F.P.); 1803085201030@stu.nchu.edu.cn (H.Z.); 1803082503025@stu.nchu.edu.cn (Y.C.)

**Keywords:** ultrasonic treatment, vacuum counter-pressure casting, microstructure, mechanical properties, ultrasonic temperature, ZL114A alloy

## Abstract

The effect of ultrasonic temperature on density, microstructure and mechanical properties of vacuum counter-pressure casting ZL114A alloy during solidification was investigated by optical microscopy (OM), scanning electron microscope (SEM) and a tensile test. The results show that compared with the traditional vacuum counter-pressure casting aluminum alloy, the primary phase and eutectic silicon of the alloy with ultrasonic treatment has been greatly refined due to the dendrites broken by ultrasonic vibration. However, the refining effect of ultrasonic treatment on vacuum counter-pressure casting aluminum alloy will be significantly affected by ultrasonic temperature. When the ultrasonic temperature increases from 680 °C to 720 °C, the primary phase is gradually refined, and the morphology of eutectic silicon also changes from coarse needle-like flakes to fine short rods. With a further increase in the ultrasonic temperature, the microstructure will coarse again. The tensile strength and elongation of vacuum counter-pressure casting ZL114A alloy increases first and then decreases with the increase of ultrasonic temperature. The optimal mechanical properties were achieved with tensile strength of 327 MPa and the elongation of 5.57% at ultrasonic temperature of 720 °C, which is 6.3% and 8.2%, respectively, higher than that of alloy without ultrasonic treatment.

## 1. Introduction

Aluminum alloys have gained wide applications in the aeronautics and spaceflight industry for their superior comprehensive properties, such as being lightweight, with excellent mechanical performance, good corrosion resistance and castability [1,2,3]. However, it is difficult to consistently produce sound castings by traditional casting methods due to the existence of microporosity and coarse microstructures [4,5] which degrade the mechanical properties of aluminum alloy castings. As an advanced counter-gravity casting technology, the vacuum counter-pressure casting method can ensure that the casting has a good solidification feeding condition, dense microstructure and excellent mechanical property due to its characteristic of filling molds under low pressure and crystallizing under high pressure [6]. Our previous research [5,6,7] indicates that there is an extrusion and infiltration effect that can promote the aluminum alloy molten metal flow toward the dendrite spaces through narrow passages during the solidification process of vacuum-counter pressure casting aluminum alloy. The dendrite will be plastically deformed, even crushed and broken under the extrusion and infiltration effect, which is beneficial to refine the grains and improve the mechanical properties of the aluminum alloy castings. 

With the application of aluminum alloys in the high-tech field, stricter requirements are imposed on the structure and properties of aluminum alloys [8,9]. In order to further improve the performance of alloys, many efforts have been made to refine the microstructure and increase the mechanical properties of aluminum alloy castings, such as electronic-magnetic stirring [10,11] and ultrasonic vibration [12,13] used in the casting process. Ultrasonic treatment (UST) is one of the most effective methods used in refining metal grains [14,15]. The cavitation effect produced by ultrasonic treatment can form lots of cavitation bubbles in the molten metal, resulting in internal evaporation of liquid and local temperature fluctuation, which are advantageous to the refinement of grains and the improvement of mechanical properties [16,17,18,19]. At present, the application of ultrasound in the process of metal-forming attracts more attention. X. Jian et al. [12] studied the effect of power ultrasound on the solidification of aluminum alloys, and the experimental results indicated that the dominant mechanism for grain refinement using ultrasonic vibration is likely not due to dendrite fragmentation but cavitation-induced heterogeneous nucleation. Tzanakis I. et al. [20,21] systematically studied the ultrasonic treatment of molten aluminum, found the ultrasonic capillary effect and considered that the ultrasonic cavitation intensity was mainly influenced by the distance from the ultrasonic source, melt temperature and input power. Cheng et al. [22] carried out the influence of high-energy ultrasonic waves on alloy melt in different melt temperatures and summarized that the melt is applied with high-energy ultrasonic at about 660 °C, which is the best for the refinement of primary silicon. 

Both the extrusion and infiltration effect during vacuum counter-pressure casting and the cavitation effect and acoustic flow effect during ultrasonic treatment are significantly affected by the melt temperature [4,23]. However, there are few studies [23] on the effect of ultrasonic temperature on the grain refinement of aluminum alloy castings, especially the effect of ultrasonic temperature on the microstructure and mechanical properties of vacuum counter-pressure casting aluminum alloys. In present studies, the ultrasonic temperature as one primary factor is selected to study its effect on density, microstructure and mechanical properties of the vacuum counter-pressure casting ZL114A alloy. The purpose of the present study is to provide theoretical basis and technical support for the application of ultrasonic melt processing technology and vacuum counter-pressure casting technology to produce high-quality large-scale complex thin-walled aluminum alloy castings. 

## 2. Materials and Methods

### 2.1. Experimental Equipment

The vacuum counter-pressure casting equipment with ultrasonic vibration is shown in Figure 1a, and it mainly includes a down kettle, upper kettle and ultrasonic vibration device. The electric resistance furnace was kept inside the down pressure kettle and the metal mold with ultrasonic device was settled in the upper pressure kettle. The ultrasonic device was composed of an ultrasonic transducer and vibration tool head. Installing the ultrasonic device on the metal mold through the flange on vibration tool head of the ultrasonic device. The working frequency of the ultrasonic wave was 19.3–20.3 kHz and the power range was 0–1000 W. Once the vacuumization stage was completed, the ultrasonic device was activated immediately until the pressure was released. The vacuum counter-pressure casting process can be divided into five stages: vacuum, filling mold with UST, rising pressure with UST, holding pressure with UST, releasing pressure, as shown in Figure 1b. 

### 2.2. Experimental Material

The experimental material was the ZL114A aluminum alloy with the chemical composition listed in Table 1.

### 2.3. Experimental Procedures

The metal mold having a cylindrical cavity with the dimension of φ 12 × 100 mm. The ZL114A aluminum alloy ingot was melted in a graphite crucible inside an electric resistance furnace. The liquid metal was refined and modified with 0.15 wt.% refining agent and 2 wt.% sodium modifier. When the temperature of molten metal reached the set ultrasonic temperature it was held for 20 min. After the upper and down tanks were vacuumed at the same time to a vacuum degree of 20 kPa, the molten metal was pressed into a metal mold cavity with a preheated temperature of 250 °C under a designed filling pressure difference of 35 kPa. At this stage, the ultrasonic vibration device was activated until the pressure was released. Four different ultrasonic temperatures (680 °C, 700 °C, 720 °C, 740 °C) were used in this paper. The temperature of the melt during vacuum counter-pressure casting with UST is called ultrasonic temperature. The difference between ultrasonic temperature and pouring temperature is whether ultrasonic treatment is applied during the experiment. Specific process parameters of vacuum counter-pressure casting are shown in Table 2.

### 2.4. Test Method

The microstructural characterization samples were taken from the castings along the direction of ultrasonic vibration and prepared by grinding paper from 800 to 2000 grit and metallographically polished with 1 μm alumina, and subsequently etched using a reagent comprising 5 mL HF and 95 mL distilled water. Metallographic observations were conducted with an XJP-6A optical microscope (OM, Jinan Zhongte testing machine Co. Ltd., Jinan, China). The eutectic silicon and fracture morphology were observed by a scanning electron microscope (SEM, Quanta 200, FEI, Brno, Czech Republic). Energy disperse spectroscopy (EDS, Oxford INCA, Virginia, UK) was used to characterize the content of the third phase of the aluminum alloy. The tensile tests were carried out using a WDW-50 testing machine (Jinan Meters Testing Technology Co., Ltd., Jinan, China) at a strain rate of 3 × 10^−3^/s according to GB/T228-2002. The sizes of the tensile specimens are shown in Figure 2. The densities of samples were measured by Archimedes’ method. Relative density is the ratio of actual density divided to theoretical density.

## 3. Results and Discussion

### 3.1. Microstructure without UST

The effect of pouring temperature on the density of vacuum counter-pressure casting aluminum alloy samples is shown in Figure 3. It can be seen from the figure that pouring temperature has a certain effect on the density of the vacuum counter-pressure casting aluminum alloy sample. Although the density of the vacuum counter-pressure casting aluminum alloy is already very high, as the pouring temperature increases from 680 °C to 740 °C, the density increases first and then decreases. When the pouring temperature is 700 °C, the density of aluminum alloy samples at all positions reaches the maximum. At this time, the relative density of sample 1 # and sample 3 # is 0.9990.

The optical microscope was used to examine the microstructure characteristic of vacuum counter-pressure casting aluminum alloys under different pouring temperature and the result is shown in Figure 4. It can be found that the primary phase of vacuum counter-pressure casting aluminum alloys changed obviously with the increase of pouring temperature. When the pouring temperature is 680 °C, the microstructure of the alloy is mainly developed dendrites cells, as shown in Figure 4a. When the pouring temperature increases to 700 °C, the number of developed dendrites in the structure is reduced, and the partial primary α-Al dendrites cell is refined, as shown in Figure 4b. Further increase the pouring temperature, and the microstructure begins to deteriorate, and many coarse primary α-Al dendrites cells appear again, as shown in Figure 4c,d. 

### 3.2. Microstructure with UST

The density is one of the critical factors that affect the mechanical property of aluminum alloys. In order to understand the effect of ultrasonic temperature on the density of vacuum counter-pressure casting ZL114A aluminum alloy, the densities of vacuum counter-pressure casting aluminum alloys under different ultrasonic temperatures were measured from the average of five measurements. The density result of vacuum counter-pressure casting alloy samples with UST is shown in Figure 5. It can be found that the effect of ultrasonic temperature on the density of vacuum counter-pressure casting aluminum alloy is very obvious, and when the ultrasonic temperature increases, the density of alloys increases first and then decreases. As the ultrasonic temperature gradually increases to 720 °C, vacuum counter-pressure casting aluminum alloy reaches the maximum density. At this time, the relative density of 1 #, 2 # and 3 # sample are 0.9999, 0.9992 and 0.9994, respectively. Compared with the samples without UST, the densities of all vacuum counter-pressure casting aluminum alloy samples with UST have been improved. 

The microstructure of vacuum counter-pressure casting aluminum alloy under different ultrasonic temperature is shown in Figure 6. It is found that ultrasonic temperature has a significant influence on microstructure of vacuum counter-pressure casting aluminum alloys. As shown in Figure 6a, when the ultrasonic temperature is 680 °C, there are no developed dendrites in the microstructure. The primary α-Al phase is mainly a rose-like dendritic cell. As the ultrasonic temperature increases to 700 °C, it can be found that small round primary α-Al phases appear in the aluminum alloy structure in Figure 6b. When the ultrasonic vibration temperature is 720 °C, the primary α-Al dendritic cells are remarkably refined and the developed dendrites are sharply reduced, as shown in Figure 6c. When the ultrasonic temperature reaches 740 °C, although the microstructure of the primary phase of the alloy can maintain a substantially round shape, the grain size is relatively coarse as shown in Figure 6d. Compared with the microstructure of vacuum counter-pressure casting aluminum alloy samples without UST, the solidification behavior of the vacuum counter-pressure casting alloy with UST has changed significantly. The primary phase is significantly refined. When the ultrasonic temperature is 720 °C, the microstructure of vacuum counter-pressure casting aluminum alloy is relatively optimal.

To further confirm the effect of ultrasonic treatment on the solidification behavior of a vacuum counter-pressure casting alloy, we observed the SEM images of the eutectic phase of alloy samples without UST and with UST, as shown in Figure 7. In the traditional vacuum counter-pressure casting alloys, although most of the eutectic silicon in the structure exists in the form of small round particles, there are still many coarse needle and strip eutectic silicon particles, as shown in Figure 7a. Compared with traditional vacuum counter-pressure casting, the eutectic silicon with UST is significantly refined, mainly in the form of small round particles, and evenly distributed, as shown in Figure 7b. This is because the silicon phase can be refined by the acoustic cavitation effect [24]. In addition, we can see the obvious third phase in Figure 7a. However, we did not see the phase from the XRD diagram. This is mainly because the volume fraction of the third phase generated is small, and the diffraction peak is not detected by X-rays. To further analyze the third phase, the third phase of 350 kPa solidification pressure was analysis by EDS, as shown in Figure 7c,d. The results show that Fe and Mg can be found in the third phase.

The SEM images of eutectic phase of vacuum counter-pressure casting aluminum alloy samples under different ultrasonic temperature is shown in Figure 8. It can be seen that ultrasonic temperature has a significant effect on the morphology of eutectic silicon. When the ultrasonic temperature is lower than 700 °C, the eutectic silicon structure in the ZL114A alloy mainly exists as thick needles or rods. In this temperature range, although increasing the ultrasonic temperature can play a certain role in refining the eutectic silicon structure, the effect is not good, as shown in Figure 8a,b. With a further increase in the ultrasonic temperature, the eutectic silicon structure is significantly refined. At 720 °C, the strip and massive eutectic silicon are converted completely into the small short rod-like and evenly distributed Si phases at the grain boundary of primary α-Al, as shown in Figure 8c. When the ultrasonic temperature continues to rise, the eutectic silicon structure begins to coarsen locally, as shown in Figure 8d. Coarse eutectic silicon phases in the structure are often detrimental to the performance of the alloys. Therefore, it is necessary to refine the eutectic silicon [25]. The application of ultrasonic vibration during the vacuum counter-pressure casting process can further refine the eutectic silicon structure, but the choice of ultrasonic temperature is very important for the refining effect. In this paper, the best refining effect is obtained at an ultrasonic temperature of 720 °C.

### 3.3. Mechanical Properties

Table 3 and Table 4 show the tensile strength and elongation of vacuum counter-pressure casting aluminum alloys without UST and with UST, which reflected the influence of UST on mechanical properties in the process of solidification. As shown in Figure 7, for traditional vacuum counter-pressure casting aluminum alloy, the tensile strength and elongation are 307.5 MPa and 5.15%, respectively, while the alloys with UST can reach 327 MPa and 5.57%. Compared with the traditional casting method, the tensile strength and elongation are increased by 6.3% and 8.2%, respectively. Both tensile strength and elongation improvement proved that ultrasonic sound could cause effective grain refinement [23,26,27]. The introduction of ultrasound can not only refine the alloy grains, but also obtain evenly distributed fine eutectic silicon phases. Therefore, vacuum counter-pressure casting aluminum alloy with UST showed better mechanical performance. The mechanical properties of the vacuum counter-pressure casting alloy is the best under an ultrasonic temperature of 720 °C. 

The study of fracture morphology further explains the influences of ultrasonic vibration on mechanical properties. Figure 9 displays the tensile fracture morphology of alloy samples both without and with UST. From Figure 9a, the fracture morphology of a vacuum counter-pressure casting sample without UST is mainly of tough dimples, and there are also some tearing ridges. By contrast with Figure 9a, the fracture morphology shows obvious ductile fracture features in Figure 9b, and the size and depth of tough dimples increase significantly. The reduction of tearing ridges and the size and depth of tough dimples increase and demonstrate great improvement of mechanical properties of vacuum counter-pressure casting alloy with UST.

### 3.4. Discussion

According to the experimental results in this paper, it can be concluded that ultrasonic treatment can significantly refine the primary and secondary phases of vacuum counter-pressure casting aluminum alloys, thereby improving the mechanical properties of aluminum alloys. However, the refinement effect of ultrasonic treatment on the microstructure of vacuum counter-pressure casting will be significantly affected by ultrasonic temperature. The optimal microstructure and mechanical properties of vacuum counter-pressure casting alloy with UST were achieved at an ultrasonic temperature of 720 °C. 

The propagation of ultrasonic waves in molten metal mainly produces two effects, namely a cavitation effect and an acoustic flow effect [28,29]. The high-pressure shock wave produced by a cavitation effect can significantly increase the melting point of the local melt, and the effective subcooling degree will also be greatly increased, thereby increasing the nucleation rate and making the grain refinement. At the same time, the collapse of acoustic cavitation bubbles produced by cavitation will produce high-speed micro-jets near the bubbles. This instantaneous high-speed acoustic flow has a stirring effect, which can disperse the broken dendrites in the molten pool of the melt, thereby increasing the number of crystal grains in the alloy melt. The specific calculation formula of the cavitation effect and acoustic flow effect produced by UST under high temperature and high pressure can be expressed by the following Equation [30]: (1)$$p_{\max}\approx p_{V}{\left[\frac{p_{m}\left(\gamma -1\right)}{p_{V}}\right]}^{\frac{\gamma }{\gamma -1}}$$
(2)$$T_{\max}\approx T_{\min}\frac{P_{m}}{P_{V}}\left(\gamma -1\right)$$
where *P*_max_ is the maximum pressure in the bubble when the transient cavitation bubble collapses; *T*_max_ is the maximum temperature in the bubble when the transient cavitation bubble collapses; *T*_min_ is the ambient temperature; *P_m_* is the total pressure of the cavitation bubble during the collapse process; *P_V_* is the vapor pressure in the cavitation bubble; *γ* is the specific heat ratio of the steam. From Equations (1) and (2), we found that a high pressure of 1 × 10^8^–1 × 10^9^ Pa and an instantaneous high temperature of 1 × 10^3^–1 × 10^4^ K will be simultaneously generated on the cavitation bubble wall with the collapse of the cavitation bubble. The course plate-like eutectic silicon that has just nucleated and grown can be easily crushed by this instantaneous high temperature and high pressure. In addition, it can be seen from Equation (1) that the high pressure generated by the collapse of the cavitation bubble will increase with the increase of external pressure, resulting in the cavitation effect and acoustic flow effect of vacuum counter-pressure casting with UST being greatly enhanced under solidification pressure. This is the main grain refinement mechanism of vacuum counter-pressure casting aluminum alloy with UST. 

Melt temperature is closely related to the cavitation effect and acoustic flow effect produced by ultrasonic vibration, and significantly affects the microstructure of the vacuum counter-pressure casting aluminum alloy. In an incompressible liquid with viscosity and surface tension, the velocity of the cavitation bubble wall can be estimated by the following Equation [31]:(3)$$U^{2}=\frac{\left(\frac{\sigma }{R}-n^{3r}P_{0}+P_{\infty }\right)\left(n^{2}-1\right)}{\frac{\rho }{2}-4\mu R\ln n}$$
where *U* is the velocity of the cavitation bubble wall; *P*_0_ is the initial vapor pressure in the bubble; *P*_∞_ is the pressure at infinity in the liquid; n is the ratio of the compressed radius of the cavitation bubble to the initial radius; *μ* is the viscosity of the liquid; *σ* is the surface tension of the liquid; *ρ* is the density of the liquid; and *R* is the radius of the bubble. It can be seen from Equation (3), ln*n* is a negative value due to *n* always being less than 1, so the viscosity term 4*μR*ln*n* will reduce the velocity of cavitation bubbles when *μ* increases. The viscosity of the molten metal will decrease when the ultrasonic temperature increases, and the movement speed of the cavitation bubble wall will increase, resulting in the growth and collapse of the cavitation bubble becoming particularly fast. The refinement effect will be strengthened. Therefore, when the ultrasonic temperature is 720, the microstructure and performance of the vacuum counter-pressure casting aluminum alloy are the best. However, as the temperature increases further, the microstructure and properties of the alloy will be deteriorated. This is because it takes a certain amount of time for ultrasonic treatment to refine the grains. When high-temperature ultrasonic is introduced, the time for ultrasonic treatment of the melt is relatively prolonged, and the thermal effect of the ultrasonic wave is enhanced, resulting in an increase in the size of the crystal grains [32]. 

## 4. Conclusions

In view of the previous results, the following conclusions can be drawn:(1)Compared with traditional vacuum counter-pressure casting aluminum alloy, the primary phase and eutectic silicon structure of the alloy with UST were drastically refined, resulting in the performance also being greatly improved. This is mainly due to the cavitation effect and acoustic flow effect of vacuum counter-pressure casting with UST being greatly enhanced under solidification pressure.(2)The effect of ultrasonic treatment on the vacuum counter-pressure casting aluminum alloy will be significantly affected by ultrasonic temperature. When the ultrasonic temperature increases, the primary phase of vacuum counter-pressure casting alloy changes significantly, from rose-like dendrites to fine equiaxed crystals. The morphology of eutectic silicon also changes from coarse needle-like flakes to fine short rods. The refinement effect will be weakened when the ultrasonic temperature is too high or too low. The best ultrasonic temperature is 720 °C.(3)The tensile strength and elongation of vacuum counter-pressure casting aluminum alloy with UST first increases and then decreases with the increase of ultrasonic temperature. The optimal mechanical properties of the alloy were achieved at an ultrasonic temperature of 720 °C. The tensile strength and elongation are 327 MPa and 5.57%, respectively. Compared with the traditional casting method, the tensile strength and elongation are increased by 6.3% and 8.2%, respectively.

## Figures and Tables

**Figure 1 materials-13-04232-f001:**
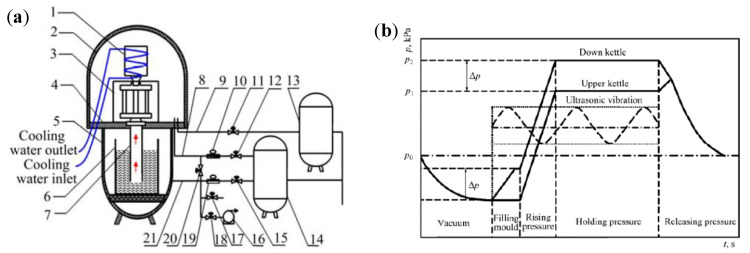
Equipment and technical schematic of synergistic action between ultrasonic vibration and vacuum counter-pressure: (**a**) equipment, 1—ultrasonic device, 2—upper kettle, 3—mould, 4—clapboard, 5—down kettle, 6—crucible, 7—rising tube, 8, 9, 21—gas tube, 10, 19—regulating valve, 11, 12, 15, 17, 18, 20—switch valve, 13, 14—gas jar, 16—vacuum pump; (**b**) technical schematic.

**Figure 2 materials-13-04232-f002:**
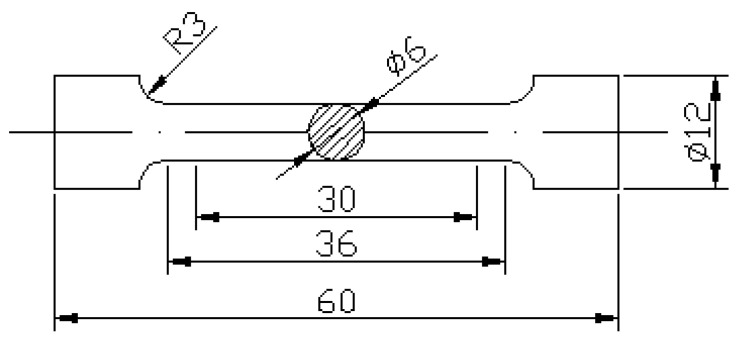
Samples for mechanical property testing (mm).

**Figure 3 materials-13-04232-f003:**
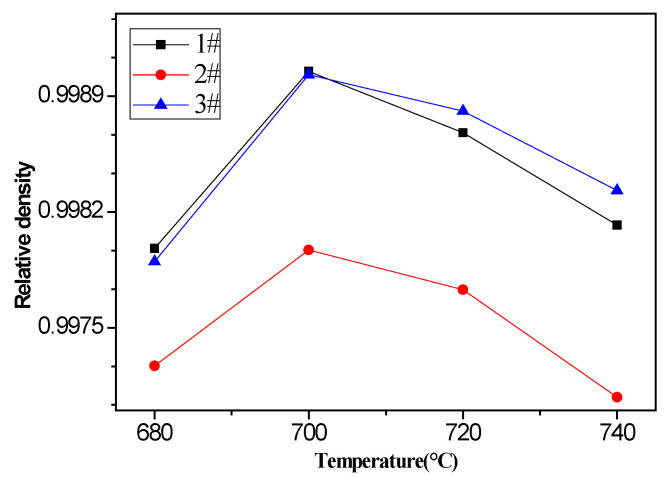
The density results of vacuum counter-pressure casting alloy samples under different pouring temperature.

**Figure 4 materials-13-04232-f004:**
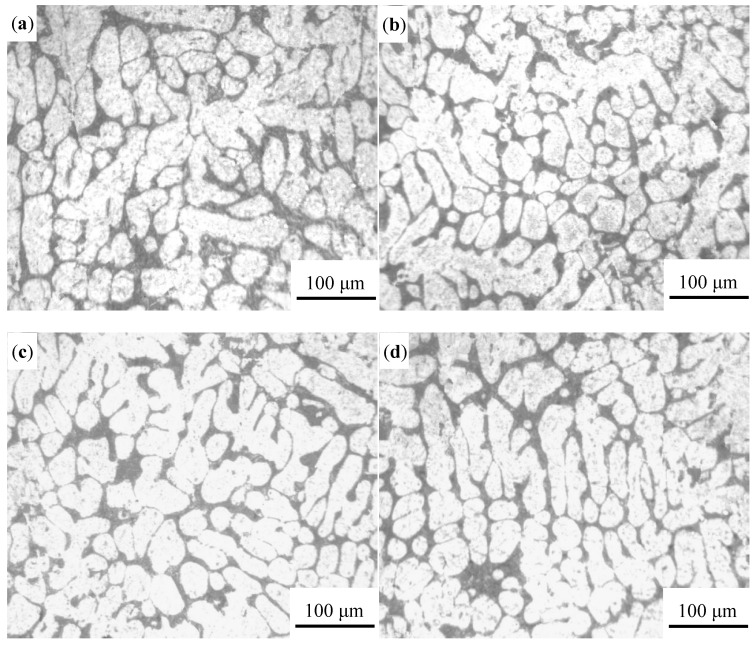
Microstructure of vacuum counter-pressure casting aluminum alloy under different pouring temperature; (**a**) 680 °C, (**b**) 700 °C, (**c**) 720 °C, (**d**) 740 °C.

**Figure 5 materials-13-04232-f005:**
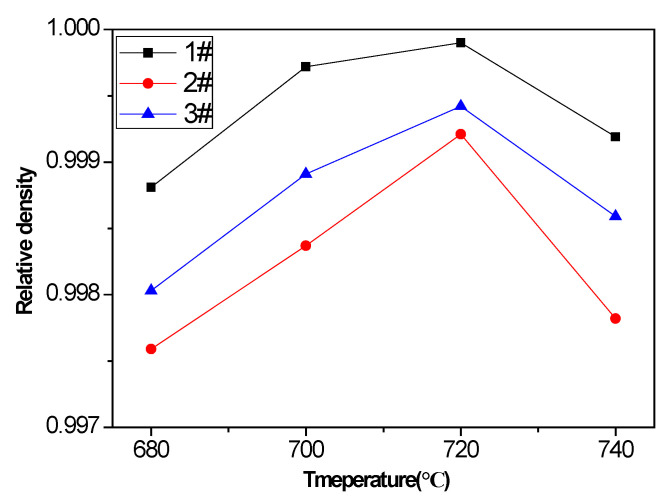
The density result of vacuum counter-pressure casting aluminum alloy sample under different ultrasonic temperature.

**Figure 6 materials-13-04232-f006:**
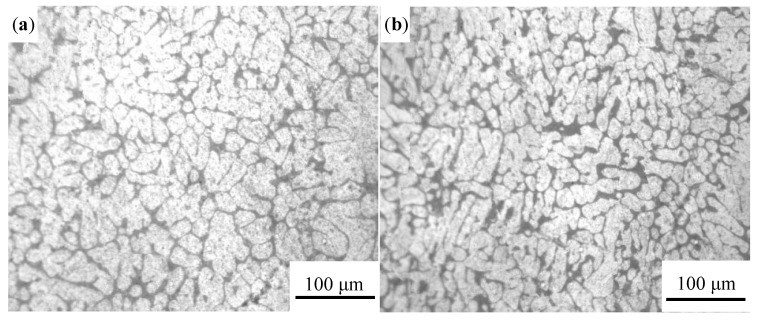
Microstructure of vacuum counter-pressure casting aluminum alloy under different ultrasonic temperature; (**a**) 680 °C, (**b**) 700 °C, (**c**) 720 °C, (**d**) 740 °C.

**Figure 7 materials-13-04232-f007:**
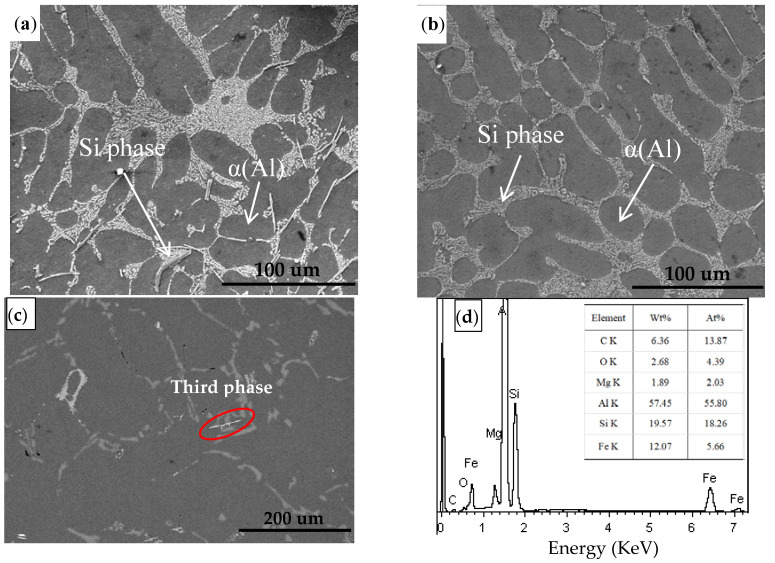
SEM image and XRD image of vacuum counter-pressure casting aluminum alloy samples without UST and with UST; (**a**) 350kPa solidification pressure, (**b**) Synergistic action of 600W ultrasonic power and 350kPa solidification pressure, (**c**) SEM image in BSE under 350 kPa solidification pressure, (**d**) Energy spectrum analysis of third phase.

**Figure 8 materials-13-04232-f008:**
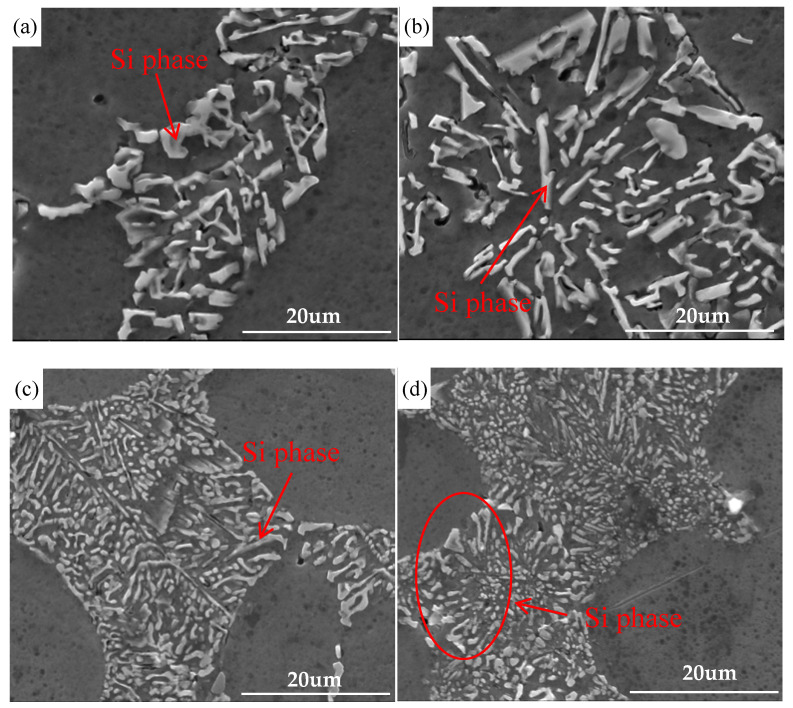
SEM images of eutectic phase of vacuum counter-pressure casting aluminum alloy under different ultrasonic temperature; (**a**) 680 °C, (**b**) 700 °C, (**c**) 720 °C, (**d**) 740 °C.

**Figure 9 materials-13-04232-f009:**
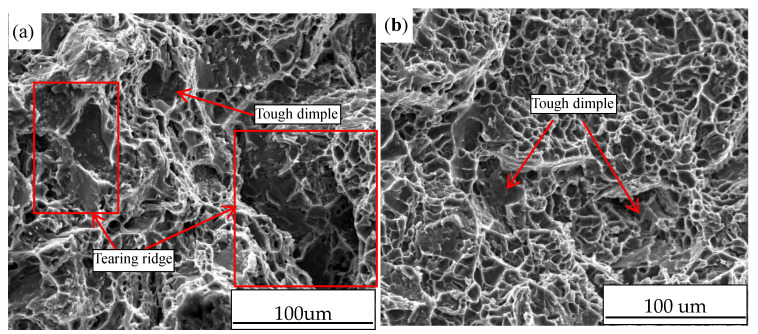
The tensile fracture morphology of vacuum counter-pressure casting samples; (**a**) without UST, (**b**) with UST.

**Table 1 materials-13-04232-t001:** Chemical composition of the ZL114A aluminum alloy (wt.%).

Si	Mg	Fe	Ti	Be	Zn	Cu	Al
6.89	0.52	0.12	0.13	0.04–0.07	<0.1	<0.01	Bal.

**Table 2 materials-13-04232-t002:** Technical parameters of vacuum counter-pressure casting with ultrasonic treatment (UST).

Ultrasonic Temperature (°C)	Vacuum Degree (kPa)	Pressure Difference (kPa)	Solidification Pressure (kPa)	Time of Holding Pressure (s)	Ultrasonic Power (W)
680	20	35	350	200	600
700					
720					
740					

**Table 3 materials-13-04232-t003:** Mechanical properties of vacuum counter-pressure casting aluminum alloy without UST.

**Temperature/°C**	680	700	720	740
**Tensile Strength/MPa**	291.9 ± 2.5	314.9 ± 2.7	307.5 ± 2.1	293.4 ± 3.3
**Elongation/%**	4.84 ± 0.13	5.38 ± 0.15	5.15 ± 0.12	4.88 ± 0.21

**Table 4 materials-13-04232-t004:** Mechanical properties of vacuum counter-pressure casting aluminum alloy with UST.

**Temperature/°C**	680	700	720	740
**Tensile Strength/MPa**	308.9 ± 3.1	318.8 ± 3.5	327 ± 4.0	310.3 ± 3.8
**Elongation/%**	4.83 ± 0.10	5.25 ± 0.15	5.57 ± 0.11	4.89 ± 0.19

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
