# Peer review of "Optimizing the Microstructure and Mechanical Properties of Vacuum Counter-Pressure Casting ZL114A Aluminum Alloy via Ultrasonic Treatment"

_materials, 2020, doi:10.3390/ma13194232_

Round 1

Reviewer 1 Report

Paper “Optimizing the microstructure and mechanical properties of vacuum counter-pressure casting ZL114A aluminum alloy via ultrasonic treatment” demonstrated an interesting results. Comments are presented below.

  1. Abstract and manuscript.

“…and the elongation of 5.57 % at ultrasonic …” 2 units after point are questionable. What is the standard deviation (error bar) of the measured value? The same for Tables 3 and 4 for tensile strength.

“ is 6.34 % and 8.16 %” 2 units should be round to 1.

  1. “The vacuum counter-pressure casting equipment with ultrasonic vibration is shown in Figure 1a, and it mainly includes down kettle, upper kettle and ultrasonic vibration device.”

Fig1 demonstrate the scheme but not the equipment.

  1. Table 1 illustrated the composition of the alloy from the standard. What was the nominal composition of the investigated alloy?
  2. Line 97. Please use the special symbol of diameter.
  3. Lines 99-100. Which refining agent was used? How 2% of sodium modifier change the chemical composition?
  4. Line 112. “metallographically polished with 1 um alumina,”. Change on special symbol.
  5. Lines 120-127 and Fig2 and 2.4. Density is usually presented in the g/cm3. Fig2 demonstrate the relative density from the process temperature. Density was measured by Archimedes’ method. But how was calculated the relative density? The same questions to Fig 4.
  6. Figure 2. What is the standard deviation (error bar) of the measured (calculated) value? Are the changes in 0.0007-0.0015 significant? The same questions to Fig 4.
  7. Microstructures in Fig 3a,b are not visible. Fig 3 demonstrate the dendrite structure and see the equiaxed crystals in the microstructure is difficult. The same for Fig 5. Grain structure was not investigated and using terms “grain” and “crystal” is not correct.
  8. Fig6a. Primary Al and Al+Si eutectic are clearly visible. And third phase (length about 20-50µm) with needle form also visible. And these needles are not presented in Fig 6b. What is the phase? Looks like Al5FeSi or Al3Fe. And what changes proceed during UST?
  9. Fig7. and Materials and Methods. Not clear – What was the pouring temperature. The morphology of the Si particles is in depends of the pouring temperature.

Reviewer 2 Report

Research focused on the field of ultrasonic melt processing is of great industrial importance. 

The article is written clearly and concisely, but I still have a few questions and comments:

1) In the "1 Introduction" section, it would be appropriate to describe in more detail the principle of the method under investigation

2) Line 65: 

Is written: "However, there are few studies on the effect of ultrasonic
66 temperature on the grain refinement of aluminum alloy castings..." 

Can you specify and describe the mentioned studies?

3) Line 108:

Table 2 - Please edit the table. The values are not clearly defined for each ultrasonic temperature

4) Line 110:

It is possible to describe in detail the experimental test methods:

  • Metallographic observation - it is possible to specify a standard for determining the structure?
  • The tensile tests - How the test rods were made? What is the shape of the test rods? Please specify a picture

Round 2

Reviewer 1 Report

Unfortunately, the previous comments still need major revisions.

Comments 1, 8. Belief alone is not enough to confirm experimentally determined values. The publication possibly uncorrected results is undesirable. All measured values should be confirmed by statistical processing.

Comment 3. ZAlSi7MglA is a marking, label but not a nominal composition. What is the concrete content of each element in the melted alloy? The range from standard in the Table is not fully informative.

Comment 7. The information about calculation of the relative density should be added in the Materials and Methods part.

 Comment 9. The most important.  Figs 3, 4,6,and7 demonstrates the dendrite structure. The dendritic cells are surrounded by binary eutectic. It is not a grains or dendrites or crystals. It is dendritic cell. The labels used are “dendrites”, “crystals”, “grains”. It is the similar terms which means grain. In the microstructure cross section we see the dendritic cells which surrounded by binary eutectic.

To see the grain structure usually used special chemical or electrochemical etching or EBSD.

Comment 10. The third phase particles are clearly seen in the structure (fig6a). Please provide the detail analyze: SEM images in BSE and Map of the alloying elements. The needles particles play an important role in the plasticity.

Round 3

Reviewer 1 Report

I do not find the point by point response, but found some of them in the manuscript. Unfortunately, the previous comments still need major revisions.

Comments 1, 8. Belief alone is not enough to confirm experimentally determined values. The publication possibly uncorrected results is undesirable. All measured values should be confirmed by statistical processing.

Comment 3. The concrete content of not each element in the melted alloy was presented. What content of the Fe? It is important to describe the phase composition of the alloy.

Comment 9. The most important.  The Author do not feel differences between basic materials science terminology: dendrite (grain) and dendritic cell (part of the grain).

Figs 3, 4,6,and7 demonstrates the dendrite structure (dendritic cell structure). No grains visible in the presented microstructure.

To see the grain structure usually used special chemical or electrochemical etching or EBSD.

Comment 10. The third phase particles are clearly seen in the structure (fig6a). Please provide the detail analyze: SEM images in BSE and Map of the alloying elements. The needles particles play an important role in the plasticity.

New comment. Not all peaks labeled in the XRD-patterns. Please use zoom and Log axes to Intensity. In present XRD: peaks in the range 35-37.5-degree, 38-43 degree and 65-77 degree (fig7b) do not named.

About possible formation of Mg2Si: 0.5%Mg is enough to Mg2Si phase formation. But this phase has another form (morphology) like “skeletal” branched crystals. It is dark phase in the OM and SEM (BSE). The same morphology have Al15(Fe,Mn)3Si2 particles – “Chinese script”.

In present images the third phase have a needle form. SEM (BSE) images are very important.
